# Impact of the Mediterranean Diet on the Gut Microbiome of a Well-Defined Cohort of Healthy Individuals

**DOI:** 10.3390/nu16060793

**Published:** 2024-03-11

**Authors:** Silvia Vázquez-Cuesta, Nuria Lozano García, Sara Rodríguez-Fernández, Ana I. Fernández-Avila, Javier Bermejo, Francisco Fernández-Avilés, Patricia Muñoz, Emilio Bouza, Elena Reigadas

**Affiliations:** 1Department of Clinical Microbiology and Infectious Diseases, Hospital General Universitario Gregorio Marañón, 28007 Madrid, Spain; silviavazquez.lab@gmail.com (S.V.-C.); nulogar@gmail.com (N.L.G.); sara.rodriguezinvestigacion@hotmail.com (S.R.-F.); emilio.bouza@gmail.com (E.B.); 2Instituto de Investigación Sanitaria Gregorio Marañón, 28007 Madrid, Spain; anaferavi@gmail.com (A.I.F.-A.); javier.bermejo@salud.madrid.org (J.B.); faviles@redcardiovascular.com (F.F.-A.); 3Biochemistry and Molecular Biology Department, School of Biology, Universidad Complutense de Madrid (UCM), 28040 Madrid, Spain; 4Department of Cardiology, Hospital General Universitario Gregorio Marañón, 28007 Madrid, Spain; 5Medicine Department, School of Medicine, Universidad Complutense de Madrid (UCM), 28040 Madrid, Spain; 6Centro de Investigación Biomédica en Red de Enfermedades Cardiovasculares (CIBERCV), 28029 Madrid, Spain; 7Centro de Investigación Biomédica en Red de Enfermedades Respiratorias (CIBERES CB06/06/0058), 28029 Madrid, Spain

**Keywords:** Mediterranean diet, microbiome, healthy cohort

## Abstract

A comprehensive understanding of gut microbiota in a clearly defined group of healthy individuals is essential when making meaningful comparisons with various diseases. The Mediterranean diet (MD), renowned for its potential health benefits, and the influence of adherence thereto on gut microbiota have become a focus of research. Our aim was to elucidate the impact of adherence to the MD on gut microbiota composition in a well-defined cohort. In this prospective study, healthy volunteers completed a questionnaire to provide demographic data, medical history, and dietary intake. Adherence was evaluated using the Med-DQI. The V4 region of the 16S rRNA gene was sequenced. Analysis of sequencing data and statistical analysis were performed using MOTHUR software and R. The study included 60 patients (51.7% females). Adherence correlated with alpha diversity, and higher values were recorded in good adherers. Good adherers had a higher abundance of *Paraprevotella* and *Bacteroides* (*p* < 0.001). Alpha diversity correlated inversely with fat intake and positively with non-starch polysaccharides (NSPs). Evenness correlated inversely with red meat intake and positively with NSPs. Predicted functional analysis highlighted metabolic pathway differences based on adherence to the MD. In conclusion, our study adds useful information on the relationship between the MD and the gut microbiome.

## 1. Introduction

The human intestinal microbiota, a complex ecosystem composed of trillions of microorganisms, plays a key role in maintaining health and preventing disease [1]. A thorough understanding of gut microbiota in a well-defined cohort of healthy individuals is crucial when making meaningful comparisons with various health or disease states [2]. Simultaneously, the Mediterranean diet (MD), typical of the Mediterranean coastal countries, has garnered attention for its association with a reduced risk of cardiovascular diseases, metabolic syndrome, type 2 diabetes (T2DM), certain types of cancer, and neurodegenerative disorders [3,4,5]. The diet’s emphasis on nutrient-rich foods, including fruits, vegetables, whole grains, and healthy fats, contributes to its anti-inflammatory and antioxidant properties, highlighting its role in maintaining general health [6,7]. Diets rich in fats and refined carbohydrates (typical Western diets) have become more widely accepted, and obesity and related metabolic disorders have become highly prevalent and are increasing significantly [8,9]. Intestinal bacteria play an important role in the synthesis and absorption of various nutrients and metabolites, including lipids, bile acids, short-chain fatty acids (SCFAs), amino acids, and vitamins.

Carbohydrates act as a source of energy, help control blood glucose and insulin metabolism, and participate in the metabolism of cholesterol and triglycerides [10]. An inadequate amount of carbohydrates in the diet can affect various physiological and metabolic processes. Highly processed foods that are rich in simple carbohydrates are less nutritious and cause a sharp rise in blood glucose compared to the complex carbohydrates found in legumes, vegetables, or whole grains [10]. Excessive consumption of simple carbohydrates can lead to obesity, which increases the risk of cardiovascular disease and T2DM. However, foods with a low glycaemic index and rich in non-starch polysaccharides protect against diabetes [10].

Dietary proteins can impact the gut’s physiological functions and influence amino acid, glucose, and lipid metabolism [11]. When consumed with carbohydrates, dietary proteins reduce glycaemic responses [12]. Some metabolic disorders, such as kwashiorkor and marasmus, are caused by severe protein deficiencies [13]. In the elderly, less severe forms of dietary protein deficiency occur, increasing their susceptibility to metabolic diseases [13]. On the other hand, excessive protein intake can compromise health, especially in those with liver or kidney dysfunction [14]. 

The role of dietary lipids in metabolic and cardiovascular health is crucial [15]. An excessive intake of long-chain saturated fatty acids (LCSFAs), together with reduced intake of unsaturated fatty acids (mainly omega-3), contributes to metabolic disorders attributed to the Western diet, obesity, and its cardiometabolic complications [15]. LCSFAs in tissues like skeletal muscle or the liver trigger metabolic inflammation and mitochondrial dysfunction, leading to metabolic derangements [15,16]. Lipotoxicity, which connects lipid metabolism to obesity and its comorbidities, may underlie the effects of LCSFAs on metabolic health [17].

Investigations comparing microbial profiles associated with diverse dietary patterns have revealed distinct signatures corresponding to varying levels of fibre intake, fat composition, and overall dietary diversity [18,19,20,21]. Mounting evidence suggests that the MD significantly influences the composition and functionality of the intestinal microbiota [22,23,24,25]. Several authors [25,26,27,28] have demonstrated the influence of the MD on gut microbial composition, as it contains specific food components and dietary habits that contribute to a healthy gut microbial community. De Filippis et al. [23] found that high adherence to the MD has a beneficial effect on the intestinal microbiota and the related metabolome. However, a critical research gap exists regarding a well-characterised cohort of healthy individuals strictly adhering to the MD. Such a cohort would provide valuable insights into the unique microbial signatures associated with this specific dietary pattern.

Studies on the influence of the MD on intestinal microorganisms require the use of validated indices to evaluate dietary adherence. Indices such as those described by Bach et al. and Gerber provide valuable tools for researchers to categorise study participants and draw meaningful conclusions [29,30,31]. The MD Score (MDS) and the PREDIMED 14-item questionnaire are examples of such tools, which enable researchers to categorise individuals based on their adherence to the MD [31,32]. 

Our primary objective was to provide data on a new, well-characterised cohort of healthy individuals and to explore discernible differences in the human gut microbiota based on their adherence to the MD.

## 2. Materials and Methods

### 2.1. Study Design and Population

This study was conducted in a tertiary hospital in Madrid, Spain (Hospital General Universitario Gregorio Marañón). Healthy volunteers were recruited from the faecal transplant clinic and comprised both donors and their relatives. The exclusion criteria were as follows: body mass index lower than 17 or higher than 30, any type of disease, including microbiota-related diseases (cholelithiasis, colorectal cancer, hepatic encephalopathy, idiopathic constipation, inflammatory bowel disease, irritable bowel syndrome, familial Mediterranean fever, gastric lymphoma or carcinoma, arthritis, asthma, atopy, dermatitis, psoriasis, autoimmune disease, fatigue syndrome, diabetes mellitus, hypercholesterolaemia, idiopathic thrombocytopenic purpura, myocardial ischaemia, metabolic syndrome, behavioural disorders, multiple sclerosis, myoclonus dystonia, non-alcoholic fatty liver disease, oxalate kidney stones, Parkinson’s disease), gastrointestinal disorders, immunologic disease, immunocompromise, alcohol intake >50 g/day, and use of antibiotics, probiotics, immunosuppressants, proton-pump inhibitors, or vaccines in the previous three months.

### 2.2. Data Collection and Classification

All participants completed a questionnaire to provide demographic information (age, sex, height, and weight), medical history, and dietary intake. Dietary intake was assessed through a validated food-frequency questionnaire [33,34].

Adherence to the Mediterranean diet was assessed using the Mediterranean dietary quality index (Med-DQI), which was first developed by Gerber et al. [31]. We classified the subjects into two groups according to the categories of good (1–4) and medium (5–9). 

Stratification by age group was as follows: children (0–2 years), teenagers (13–18 years), young adults (19–30 years), middle-aged adults (31–48 years), and older adults (49–76 years).

### 2.3. Microbiome-Related Definitions

The number of distinct species found in a sample was our measure of richness [35]. Evenness, as determined by the Pielou index, is the degree to which the abundance of various species is uniform or similar. Diversity indicates the degree of species richness and abundance, where alpha diversity refers to the diversity within an individual. We also applied the inverse Simpson index, which is an indicator of richness in a community with the same evenness, and the Shannon index, which takes into account the number of species living in a habitat (richness) and their relative abundance (evenness). Beta diversity refers to the difference in diversity between individuals [35].

### 2.4. Sample Processing

Immediately after receipt, the stool samples were homogenised, aliquoted, and stored at −80 °C. Total DNA was isolated from faeces employing the Fast QiaAmp DNA stool mini-kit (QIAGEN, Valencia, CA, USA) according to the manufacturer’s instructions, with the inclusion of a physical lysis step. Faeces were lysed twice for 45 s at 6.5 m/s in a FastPrep-24 instrument (MPBio, Derby, UK) with lysis matrix E tubes (MPBio, Derby, UK). The hypervariable V4 region of the 16S rRNA gene was amplified using PCR, with the following primers: 515: GTGCCAGCMGCCGCGGTAA and 806: GGACTACHVGGGTWTCTAAT, tailed with sequences to incorporate Illumina flow-cell adapters and indexing barcodes (Illumina, San Diego, CA, USA). 

Agencourt Ampure Beads (Beckman Coulter, Barcelona, Spain) were used to remove primer dimers and low-molecular-weight products. All samples were quantified, and their quality was evaluated for amplicon size with 4200 TapeStation (Agilent Technologies, Santa Clara, CA, USA). Finally, amplicons were pooled equimolarly and sequenced (2 × 250) on an Illumina Miseq platform (Illumina, San Diego, CA, USA) according to standard protocols.

### 2.5. Data Analysis

Raw data preprocessing, classification by operational taxonomic units (OTUs) with 97% similarity, and taxonomical classification were performed using MOTHUR software (Patrick D. Schloss, PhD, © 2024, Michigan, USA) [36] and the SILVA and RDP databases. We used MOTHUR v1.46.0 and R software v4.3.1 (R Core Team, 2021, Vienna, Austria) [37] for the analyses of beta diversity (Bray–Curtis distance, un/weighted unifrac distance), species richness (number of OTUs observed), evenness (Pielou index), and alpha diversity (Shannon index). 

Enterotype analysis was performed using the www.enterotypes.org web tool. The classification was made based on the original enterotype definition described by M. Arumugam et al. [38].

Statistical analyses were carried out using R (R Core Team, 2021, Vienna, Austria) [37]. For qualitative variables, frequencies and proportions were calculated. Regarding quantitative variables, the median and interquartile range (IQR) or mean and standard deviation (SD) were calculated. Microbiota analyses were performed using R with the packages phyloseq, microbiome, microbiomeStat, vegan, DESeq2, and microeco.

Maaslin2 was used to study the correlation between nutrient intake and taxonomy abundance based on a negative binomial model with the cumulative sum scaling normalisation method. Phylogenetic Investigation of Communities by Reconstruction of Unobserved States (PICRUSt2) was applied to identify differences in 16S rRNA-based functional prediction of the metagenome and Kyoto Encyclopedia of Genes and Genomes (KEGG) functional orthologs [39]. That is, PICRUSt2 is used to predict the functional potential of a bacterial community.

The χ^2^ test was employed to evaluate the differences between the groups. Continuous variables were compared using the *t* test or the Mann–Whitney test (when a normal distribution could not be assumed). The Kolmogorov–Smirnov test with Lilliefors correction was used to assess the normality distribution of the continuous variable. 

## 3. Results

### 3.1. Age and Sex

We enrolled 60 subjects, of whom 51.7% (31/60) were females. The median age was 31 years (IQR 24.00–49.75) (Table 1). We found significant differences in richness, alpha diversity, and evenness between females and males, with higher values among females for richness, the Shannon index, the inverse Simpson index, and the Pielou index (Figure 1).

Regarding age, we classified patients into five groups (children, teenagers, young adults, middle-aged adults, and older adults). The children’s group consistently exhibited the lowest values across all the diversity indexes. Significant differences in alpha diversity and evenness were noted, with the teenager group showing higher richness values than the other age groups and the older adult group exhibiting higher Shannon, inverse Simpson, and Pielou index values than the other groups (Figure 2).

Further analysis, stratifying by sex, indicated that these differences persisted in males but not in females, except for evenness, which remained significant in females. In terms of beta diversity, no significant differences were observed according to sex, although significant differences were found between age groups, with the children’s group being the furthest away from the rest.

Concerning the composition of the gut microbiota, Firmicutes was the most abundant phylum in both males and females, followed by Bacteroidetes and Actinobacteria. When separated by age, the children’s group had the highest abundance of Firmicutes (40.9%), followed by Actinobacteria (27.5%) and Proteobacteria (16.8%). In teenagers, Firmicutes (55.8%) predominated, followed by Actinobacteria (21%) and Bacteroidetes (18.6%). In the remaining groups, the most abundant phyla were Firmicutes (54–60%), Bacteroidetes (24–34%), and Actinobacteria (9–11%) (Figure 3).

The most abundant genera in each age group were *Bifidobacterium* (27.2%) and *Bacteroides* (12%) in children; *Bacteroides* (11.36%) and *Bifidobacterium* (10. 33%) in teenagers; *Bacteroides* (21.6%) and *Faecalibacterium* (7.3%) in young adults; *Bacteroides* (14.6%) and *Prevotella* (8.8%) in middle-aged participants; and *Bacteroides* (16.15%) and *Blautia* (7.6%) in older adults.

The distribution of enterotypes did not significantly differ between age groups or between the sexes according to our findings.

### 3.2. Adherence to the Mediterranean Diet

Of the 39 subjects who completed the food-frequency questionnaire, 21 were females and 18 were males. The median age was 34 years (IQR 26–54.5). Individuals were classified according to the quality of their adherence to the Mediterranean diet using the Med-DQI. The median obtained for the Med-DQI was 5 (IQR 4–7) (Table 2).

We classified the subjects into two groups according to their adherence to the Mediterranean diet: good adherence and medium. The good-adherence group comprised 12 females and 5 males. The median age was 48 years (IQR 28–55). The medium-adherence group comprised 9 females and 13 males, and the median age was 30 years (IQR 26–49).

Significant differences in alpha diversity were found between these groups, with a higher Shannon index and a higher inverse Simpson’s index in the good-adherence group (Figure 4). No significant differences were observed in beta diversity. There was no difference in the distribution of enterotypes between the two groups.

Subjects with good adherence to the Mediterranean diet had a higher abundance of *Paraprevotella* and *Bacteroides* (*p* < 0.001).

Analysis of adherence by sex evidenced no significant differences in alpha or beta diversity. In terms of genus abundance, we found that females with medium adherence to the Mediterranean diet had a lower abundance of *Methanosphaera*, *Paraprevotella*, *Catenibacterium*, *Collinsella*, *Clostridium_IV*, *Faecalibacterium*, and some OTUs of *Bacteroides* and *Phascolarctobacterium*, although the last two genera also had OTUs with higher abundance than females with good adherence to the Mediterranean diet (all *p* < 0.05). Among male patients, we found that those with poorer adherence to the Mediterranean diet had lower abundance of *Olsenella*, *Butyricicoccus*, *Bacteroides*, *Prevotella*, and *Paraprevotella* than those with better adherence (all *p* < 0.05) (Figure 5).

When we compared subjects with good adherence to the Mediterranean diet within each age group with those with medium adherence, we found the following genera with significant changes in abundance: In the Teenager group, we found that individuals with good adherence had decreased quantities of *Acidaminococcus*, *Cerasicoccus*, *Barnesiella*, *Butyricicoccus*, *Holdemanella*, *Paraprevotella*, and *Streptococcus*. In the young adult group, *Ruminococcus*, *Butyricicoccus*, *Clostridium IV*, and *Phascolarctobacterium* were increased in the good-adherence group. In the middle-aged group, more genera appeared, with significant changes according to diet type. Individuals with good adherence had fewer *Succiniclastum*, *Methanobrevibacter*, *Paraprevotella*, and *Megasphaera*. The abundance of the following genera increased: *Mogibacterium*, *Alloprevotella*, and *Megamonas*. We found several genera in which some OTUs increased and others decreased within the same genus, including *Prevotella*, *Bacteroides*, and *Phascolarctobacterium*. Finally, among older adults, we found that the only genera that increased with good adherence were *Collinsella* and one OTU of *Prevotella*, whereas *Bacteroides* and three OTUs of *Prevotella*, *Coprococcus*, *Akkermansia*, and *Elusimicrobium* decreased.

When we analysed alpha diversity in relation to the intake of different nutrients or food groups, we found that the Shannon index correlated inversely with fat intake and positively with non-starch polysaccharides (NSPs) and total sugars. With respect to the Pielou index, we found a negative correlation between this index and red meat intake and a positive correlation with NSPs and total sugars (all *p* < 0.05).

The correlation between nutrient intake and abundance of bacterial genera correlated positively with the consumption of olive oil and the abundance of *Odoribacter*, *Clostridium XIVb*, *Victivallis*, *Bilophila*, *Dialister*, and *Phascolarctobacterium* and negatively with *Lactococcus*, *Faecalicoccus*, *Slackia*, *Clostridium sensu stricto*, *Romboutsia*, and *Collinsella*, among others (all *p* < 0.05).

For NSP, we found positive correlations with *Finegoldia*, *Lactococcus*, *Peptoniphilus*, *Victivallis*, *Anaerofustis*, *Senegalimassilia*, and *Phascolarctobacterium* and negative correlations with *Olsenella*, *Dialister*, *Parvimonas*, and *Bifidobacterium* (all *p* < 0.05).

Positive correlations with AOAC fibre were observed for *Phascolarctobacterium* and *Dialister*; negative correlations were observed for *Olsenella*, *Anaerofustis*, *Senegalimassilia*, *Victivallis*, *Peptoniphilus*, *Slackia*, *Lactococcus*, and *Finegoldia* (all *p* < 0.05).

Glucose intake correlated positively with *Faecalicoccus*, *Clostridium sensu stricto*, *Butyricimonas*, and *Dialister* and negatively with *Victivallis*, *Senegalimassilia*, *Phascolarctobacterium*, and *Olsenella* (all *p* < 0.05).

Red meat intake correlated positively with *Faecalicoccus*, *Subdoligranulum*, *Clostridium sensu stricto*, *Anaerofustis*, and *Phascolarctobacterium* and negatively with *Dialister*, *Senegalimassilia*, *Coprobacter*, *Olsenella*, and *Intestinimonas* (all *p* < 0.05). Finally, for vegetable and fruit intake, positive correlations were recorded with *Olsenella*, *Dialister*, and *Christensenella* and negative correlations were recorded with *Faecalicoccus*, *Finegoldia*, *Butyricimonas*, and *Peptoniphilis*, among others (all *p* < 0.05) (Figure 6). 

### 3.3. Predicted Functional Metagenome Analysis

We used PICRUSt2 to identify differences in 16S rRNA-based functional predictions of the metagenome and KEGG functional orthologs. Subjects with medium adherence to the Mediterranean diet exhibited a higher abundance of the ko05219 bladder cancer pathway and overexpression of the ko00624 polycyclic aromatic hydrocarbon degradation pathway. The ko04962 vasopressin-regulated water reabsorption pathway was overexpressed in individuals with good adherence to the Mediterranean diet compared with those with medium adherence (Figure 7).

In participants who adhered to the Mediterranean diet more closely, we found a greater abundance of enzymes involved in the lipid metabolism (glycerol-3-phosphate oxidase, Octanoyl-[GcvH]:protein N-octanoyltransferase, and malonyl-S-ACP decarboxylase), those that use malonic acid as a carbon source for growth (acetyl-S-ACP:malonate ACP transferase and malonate decarboxylase holo-[acyl-carrier protein] synthase). We also found a greater abundance of enzymes in secondary metabolite biosynthesis (arogenate dehydrogenase (NADP(+)) and IgA-specific serine endopeptidase), aminoacyl-tRNA biosynthesis (O-phospho-L-seryl-tRNA:Cys-tRNA synthase and O-phosphoserine--tRNA ligase), and synthesis of bacterial lipopolysaccharides (3-deoxy-D-manno-octulosonic acid kinase).

Analysis of metabolic pathways in healthy subjects showed those with good adherence to the Mediterranean diet to be characterised by greater representation of two biosynthesis pathways, namely, the superpathway of (Kdo)2-lipid A biosynthesis and ppGpp biosynthesis and two toluene degradation pathways, namely, toluene degradation I (aerobic) (via o-cresol) and toluene degradation II (aerobic) (via 4-methylcatechol). Subjects with poorer adherence to the Mediterranean diet were characterised by greater representation of the following: biosynthesis pathways such as protein N-glycosylation (bacterial), CMP-pseudaminate biosynthesis, and the superpathway of demethylmenaquinol-6 biosynthesis II; degradation pathways such as L-histidine degradation II, L-valine degradation I, and the superpathway of pyrimidine ribonucleosides degradation; and generation of precursor metabolites and energy pathways such as ethylmalonyl-CoA pathway, glycolysis V (Pyrococcus), and L-glutamate degradation VIII (to propanoate) (Figure 8).

## 4. Discussion

We assessed the influence of the Mediterranean diet on the intestinal microbiome in a rigorously defined cohort of 60 healthy subjects classified based on age and sex. We assessed subjects’ adherence to the Mediterranean diet using the Med-DQI. This research revealed significant differences in alpha diversity, beta diversity, and microbial composition associated with adherence to diet. Notably, individuals with better adherence displayed a higher abundance of specific bacterial taxa and functional pathways. The analysis also extended to the influence of age and sex on the microbiome variations observed.

Our findings align with those of previous studies documenting the influence of the Mediterranean diet on gut microbiota diversity. The higher values recorded for richness, the Shannon index, the inverse Simpson index, and the Pielou index in the good-adherence group are consistent with the literature, thus highlighting the positive impact of this diet on microbial diversity [7,22] and emphasising the robustness of our results in reinforcing the link between the Mediterranean diet and a more diverse gut microbiome.

Exploration of the effects of age and sex on the gut microbiome sets our study apart. Although previous research has acknowledged the role of these factors, the comprehensive classification into five age groups and the separation by sex allowed for a more refined understanding of variations in microbiomes. Notably, the persistence of certain differences in males but not in females, especially in the context of evenness, raises intriguing questions that warrant further exploration.

As we can observe in our data, there was a change in most phyla between children and other age groups. This is determined because in infants, *Bifidobacterium* are abundant, and as complementary feeding is introduced, other microorganisms, such as Firmicutes and *Prevotella*, increase due to the introduction of foods rich in fibre and carbohydrates and *Bacteroidetes* due to the introduction of animal proteins [40].

Within Firmicutes, the most common families in our volunteers were *Lachnospiraceae* and *Ruminococcaceae*. These Firmicutes families hydrolyse starch and other sugars and produce butyrate and other SCFAs [41]. SCFA activity modulates the microbial environment and directly interacts with the host immune system [42]. In addition, SCFAs lead to changes in glycolysis and fatty acid metabolism in colonic epithelial cells and a decrease in inflammatory markers [43]. Within Firmicutes, elevated abundance of *Lachnospiraceae* was positively correlated with glucose and/or lipid metabolism [44,45].

Certain species respond to certain dietary carbohydrate changes, mainly bacteria specializing in using resistant starch or non-starch polysaccharides (NSPs). Some members of the *Roseburia* group were major responders to diets enriched in resistant starch [46]. Some studies have shown that a diet rich in whole-grain cereals increases the alpha diversity and abundance of Firmicutes [47]. Other *Lachnospiraceae* species are strongly influenced by NSP-rich diets [43]. However, some genera of *Lachnospiraceae* actively alter glucose metabolism, leading to inflammation and promoting the development of both type 1 and type 2 diabetes [48,49,50]. Different species of *Lachnospiraceae* have been associated with impaired lipid metabolism and, thus, obesity. It should not be forgotten that certain dietary fats, such as omega-3 polyunsaturated fatty acids, can reduce the risk of death from coronary heart disease and the development of breast cancer [51,52]. In animal models, a diet enriched with omega-3 has been shown to increase the abundance of *Lachnospiraceae* [53].

In contrast, the *Ruminococcaceae* family is strictly anaerobic and present in the colonic mucosa of healthy individuals [54]. In inflammatory bowel diseases, hepatic encephalopathy, and *Clostridioides difficile* infections, a decreased abundance of *Ruminococcaceae* has been observed [55,56,57]. This family produces butyrate and other SCFAs and therefore plays an important role in maintaining gut health.

In concordance with the literature [23,25,30], our study demonstrated a positive correlation between adherence to the Mediterranean diet and the abundance of health-associated genera such as *Paraprevotella* and *Bacteroides*. However, detailed analysis by sex uncovered updated differences, with females exhibiting specific variations in microbial abundance that were not evident in males. This suggests a potential sex-specific response to dietary patterns, an aspect that has been underexplored in the literature.

*Paraprevotella* generates succinic acid and acetic acid as end products of metabolism [58], and *Bacteroides* ferment undigested carbohydrates, producing short-chain fatty acids as end products [59]. We found several microorganisms (e.g., *Victivallis*, *Intestinimonas*, and *Olsenella*) to be directly correlated with the consumption of foods related to better health, such as olive oil, vegetables, and fruits. These microorganisms have been linked to a healthy response to various diseases [60,61]. In addition, other bacterial genera, such as *Faecalicoccus* or *Clostridium sensu stricto*, which had previously been associated with diseases such as immune-mediated inflammatory disease or inflammation [62,63], were positively correlated with glucose and red meat consumption and inversely correlated with olive oil, vegetable, and fruit consumption.

The positive correlation we found between Shannon’s index (alpha diversity) and NSPs was also observed by Martinez et al., who fed mice a diet rich in NSPs [47]. The effects of a Western diet that is rich in fats and simple sugars on gut microbiota usually involve a decrease in alpha diversity [64]. We found the same correlation with respect to fats but not with respect to total sugars. Note that our group of volunteers followed a Mediterranean diet, and none had poor adherence; therefore, sugar consumption was not very high in any case.

In relation to the correlation of some nutrients with specific genera, we can observe that in the study of olive oil consumption, we found a positive correlation with genera that have been seen to be found with a lower abundance in certain diseases such as inflammatory bowel disease, hypercholesterolaemia, gastric cancer, or type II diabetes mellitus [19,55,65,66,67,68,69]. This suggests that these microorganisms are markers of good health status. In turn, the genera we found with negative correlations with olive oil consumption were related to inflammatory bowel disease, immune-mediated diseases, and obesity [61,62,63,70,71,72,73].

In relation to the intake of NSPs, we found a positive correlation of genera such as *Lactococcus*, *Victivallis*, *Anaerofustis*, *Senegalimassilia*, and *Phascolarctobacterium*, which are related to SCFA production, good health status, and higher longevity [63,67,69,74]. However, *Finegoldia* and *Peptoniphilus* are also associated with inflammatory processes and the presence of conventional adenomas [75,76].

The genera *Olsenella* and *Parvimonas* were negatively correlated with NSPs and are associated with osteoporosis, markers of inflammation, ulcerative colitis, and colorectal cancer [77,78,79]. However, we also found negative correlations with *Dialister*, which is decreased in patients with gastric cancer [68], and with *Bifidobacterium*, although in this case, it has been shown that carbohydrates are essential for the colonisation of this genus and there is a great variety of species depending on the diet or the type of carbohydrates consumed [80]. We found positive correlations of the genera *Faecalicoccus*, *Clostridium sensu stricto*, *Butyricimonas*, and *Dialister* with glucose levels. The first three have been related to inflammatory bowel disease and cirrhotic patients with hepatocellular carcinoma [63,81]; however, *Dialister* is decreased in patients with gastric cancer [68]. With respect to a negative correlation with glucose, we found genera related to good health or even protective against some diseases, such as pancreatic cancer [67,82], but we also found *Olsenella*, which is more related to an increase in diseases such as ulcerative colitis [78].

Finally, concerning the genera that correlated positively with red meat consumption, we found genera related to inflammatory bowel disease, such as *Faecalicoccus* and *Clostridium sensu stricto* [62,63]; *Anaerofustis*, which is related to ulcerative colitis but also to greater longevity [63,83]; and other genera that are more related to a good state of health, such as *Phascolarctobacterium* and *Subdoligranulum* [69,84]. The genera found to be inversely correlated with red meat consumption were mostly related to anti-inflammatory function, protection against some types of cancer, or low risk of celiac disease [82,85,86].

The integration of functional metagenomic analysis using PICRUSt2 v 2.5.2 adds a layer of depth to our study. While previous studies have explored taxonomic composition, functional insights into metabolic pathways provide a more holistic understanding. The overexpression of specific pathways in subjects with good adherence, such as the superpathway of (Kdo)2-lipid A biosynthesis and ppGpp biosynthesis, is a key pathway in several central functions of bacteria, as well as in adaptation and resistance processes [87,88]. Toluene degradation pathways are related to the transformation of toxic aromatic hydrocarbons into other compounds that are less toxic to the host and cause less inflammation [89].

We found a greater representation of enzymes involved in lipid metabolism in individuals with good adherence to the Mediterranean diet. It is well known that the metabolism of complex lipids by the intestinal microbiota modulates the lipid homeostasis of the host, and that a lipid imbalance can have important consequences on health [90]. To date, the exact relationship of these particular pathways has not yet been explored. Other enzymes linked to Mediterranean diet adherence in our study have been found to be related to the metabolism of L-tyrosine, which is a precursor of some neurotransmitters, such as adrenaline and dopamine, and has important effects on behaviour or mood [91]. Owing to the importance of this amino acid, its depletion in the diet has been associated with an increased risk of clinical depression [91].

Notably, the absence of this level of functional analysis in many studies underscores the novel contribution of our research.

Despite the strengths of our research, some limitations must be acknowledged. The cohort’s relatively small size may limit the generalisability of our findings. In addition, the cross-sectional design precludes the establishment of causation. Future longitudinal studies with larger cohorts could provide a more robust understanding of the dynamics of the Mediterranean diet, the gut microbiome, and health outcomes. The analyses of individual nutrients should be taken as informative, since a nutrient cannot be considered in isolation but is affected by various factors.

## 5. Conclusions

In conclusion, our study provides helpful insights into the complex relationship between the Mediterranean diet and the intestinal microbiome. Thorough classification by age and sex, along with the integration of functional metagenomics, distinguishes our research from the existing literature. In this study, we observed the evolution of the gut microbiota of healthy individuals with age and were able to distinguish changes in this microbiota within a Mediterranean diet based on adherence. Thus, we found that individuals with good adherence to the Mediterranean diet had a higher abundance of genera such as *Paraprevotella* and *Bacteroides*, which are associated with good health. We also observed a correlation between certain health-promoting foods, such as olive oil or fibre consumption, and certain bacterial genera that are related to the synthesis of SCFAs and the absence of disease. Within the Mediterranean diet, we observed certain foods, such as red meat, that are associated with microorganisms that are less beneficial to health.

The positive correlations between adherence and microbial diversity and the identification of specific taxa and functional pathways provide further insight. These findings not only affirm the existing knowledge but also open avenues for extending research into sex-specific responses and the functional implications of dietary patterns on the gut microbiome. Ultimately, our study adds depth to the understanding of how dietary choices shape the microbial landscape and consequently influence human health.

## Figures and Tables

**Figure 1 nutrients-16-00793-f001:**
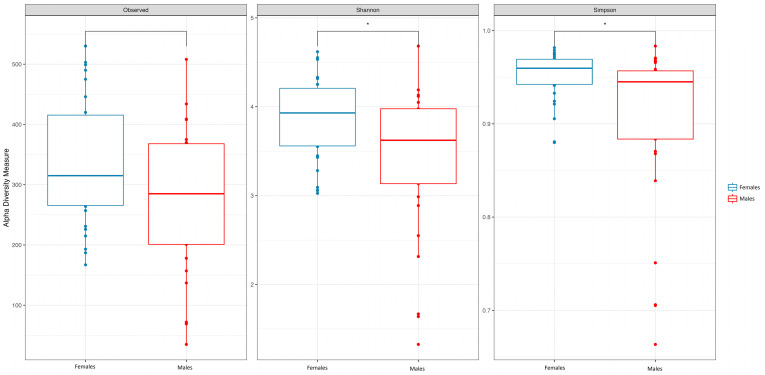
Box plot of Shannon, Simpson, and observed indexes of females and males. *, *p* <= 0.05.

**Figure 2 nutrients-16-00793-f002:**
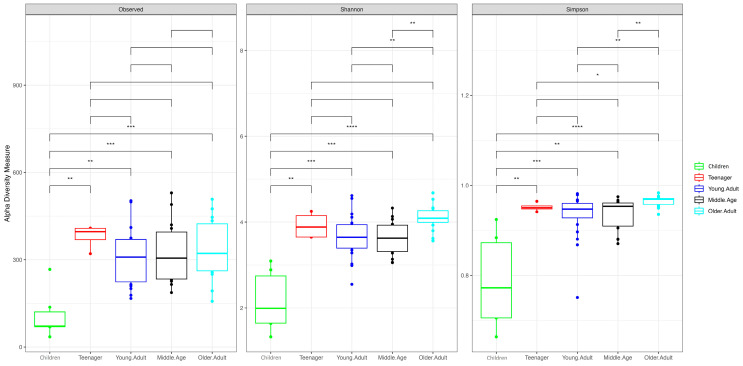
Box plot of Shannon, Simpson, and observed indexes of subjects by age. *, *p* <= 0.05; **, *p* <= 0.01; ***, *p* <= 0.001; ****, *p* <= 0.0001.

**Figure 3 nutrients-16-00793-f003:**
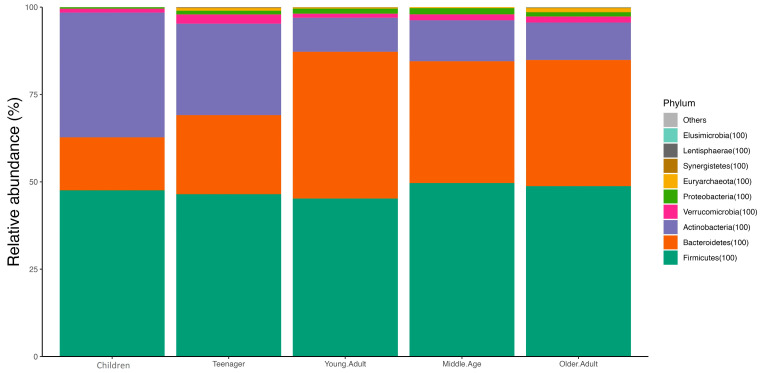
Relative abundance of phyla by age.

**Figure 4 nutrients-16-00793-f004:**
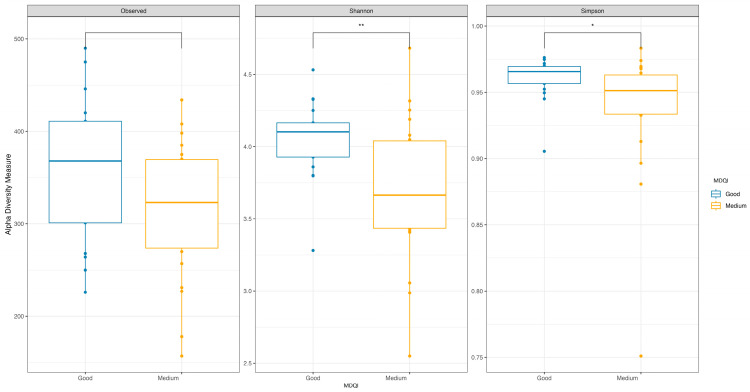
Box plot of alpha diversity index (Shannon and Simpson) and richness index (observed) values of subjects by Mediterranean diet adherence. *, *p* <= 0.05; **, *p* <= 0.01.

**Figure 5 nutrients-16-00793-f005:**
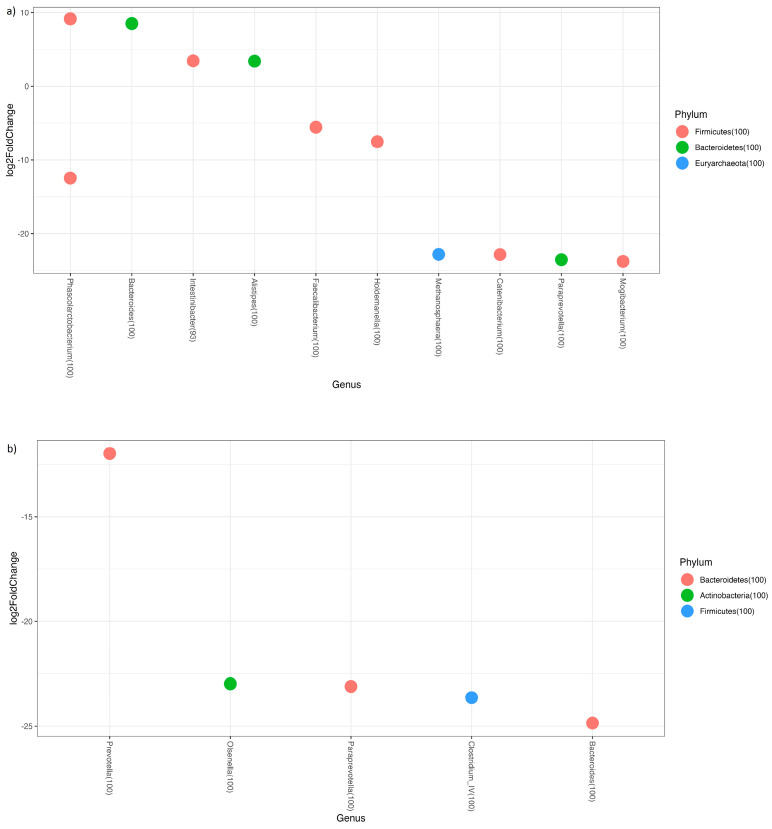
Genera Log2 fold changes representing differential abundance of genera. (**a**) Females with medium adherence to the Mediterranean diet vs. good adherence. (**b**) Males with medium adherence to the Mediterranean diet vs. good adherence.

**Figure 6 nutrients-16-00793-f006:**
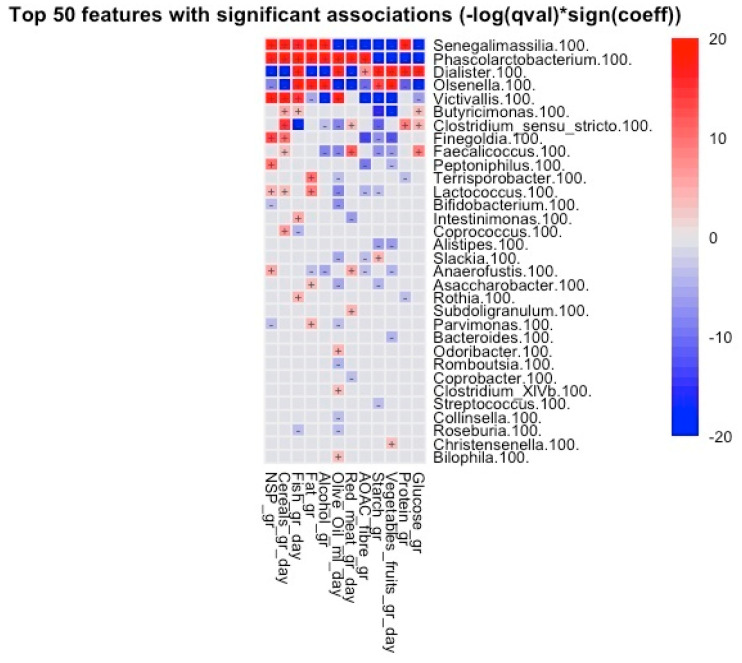
Heatmap showing the correlations between the intake of nutrients and the abundance of bacterial genera. “+” is more positive correlation: “−” is negative correlation.

**Figure 7 nutrients-16-00793-f007:**
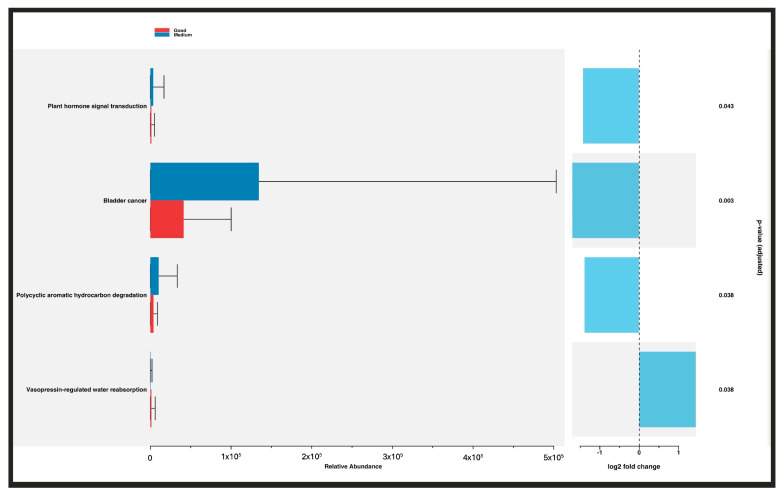
Differences in the abundance of KEGG pathways in relation to adherence to the Mediterranean diet.

**Figure 8 nutrients-16-00793-f008:**
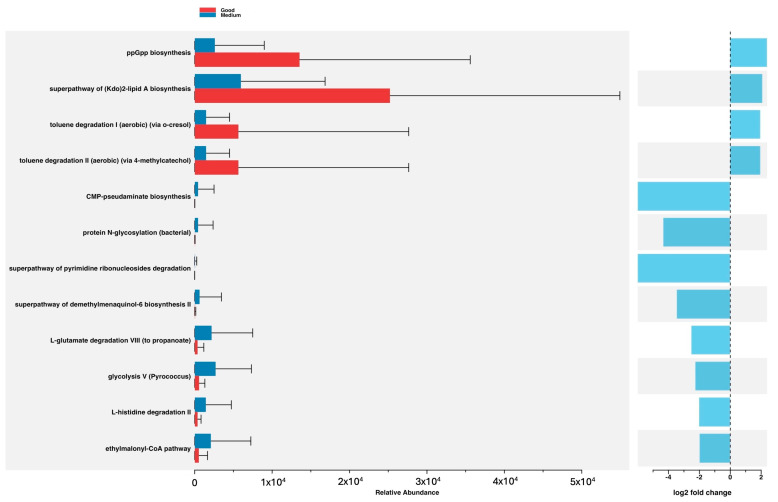
Differences in the abundance of metaCyc pathways in relation to adherence to the Mediterranean diet.

**Table 1 nutrients-16-00793-t001:** Age of participants, diversity index, and distribution of enterotypes by sex.

	FEMALES (N 31)	MALES (N 29)	TOTAL (N 60)	*p*-Value
AGE Median (IQR)	37.00 (28.50, 54.50)	26.00 (19.00, 36.00)	31.00 (24.00, 49.75)	0.006
Diversity Index				
SOBS Median (IQR)	318.55 (261.16, 409.64)	278.66 (193.37, 366.72)	312.02 (225.66, 381.65)	0.052
Inv.Simpson Median (IQR)	24.74 (17.37, 32.50)	17.62 (8.57, 23.59)	21.04 (13.03, 30.21)	0.015
Shannon Median (IQR)	3.93 (3.55, 4.20)	3.63 (3.14, 3.94)	3.74 (3.38, 4.10)	0.023
Pielou Median (IQR)	0.68 (0.62, 0.71)	0.64 (0.58, 0.68)	0.67 (0.61, 0.70)	0.033
Enterotype (ET)				0.334
N-Miss	1	5	6	
ET_Bacteroides	16 (53.3%)	9 (37.5%)	25 (46.3%)	
ET_Firmicutes	6 (20.0%)	4 (16.7%)	10 (18.5%)	
ET_Prevotella	8 (26.7%)	11 (45.8%)	19 (35.2%)	

Differences in age, diversity index, and enterotype distribution by sex.

**Table 2 nutrients-16-00793-t002:** Demographic data, Med-DQI values, and enterotype distribution by adherence to the Mediterranean diet.

	GOOD ADHERENCE (N 17)	MEDIUM ADHERENCE (N 22)	TOTAL (N 39)	*p*-Value
Age Median (IQR)	48.00 (28.00, 55.00)	30.00 (26.00, 48.75)	34.00 (26.00, 54.50)	0.122
Sex				0.065
Females	12 (70.6%)	9 (40.9%)	21 (53.8%)	
Males	5 (29.4%)	13 (59.1%)	18 (46.2%)	
Med-DQI Median (IQR)	4.00 (3.00, 4.00)	7.00 (6.00, 7.00)	5.00 (4.00, 7.00)	<0.001
Enterotypes (ETs)				0.945
ET_Bacteroides	6 (35.3%)	8 (36.4%)	14 (35.9%)	
ET_Firmicutes	4 (23.5%)	6 (27.3%)	10 (25.6%)	
ET_Prevotella	7 (41.2%)	8 (36.4%)	15 (38.5%)	

Age, sex, Mediterranean dietary quality index (Med-DQI), and enterotype distribution by adherence to the Mediterranean diet.

## Data Availability

The study’s data have been uploaded to the European Nucleotide Archive (ENA) at EMBL-EBI under accession number PRJEB57947. (https://www.ebi.ac.uk/ena/browser/view/PRJEB57947) (Accessed on 29 December 2022).

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
