# Peer review of "Impact of the Mediterranean Diet on the Gut Microbiome of a Well-Defined Cohort of Healthy Individuals"

_nutrients, 2024, doi:10.3390/nu16060793_

Round 1
Reviewer 1 Report
Comments and Suggestions for Authors
This is a cross-sectional study of the microbiota in a group of Spanish fecal transplant donors and their relatives. It asked subjects to document their adherence to a mediterranean diet using the Med-DQI. Adherence was associated with greater alpha-diversity and increased counts of Paraprevotella and Bacteriodes. Alpha diversity was negatively correlated with fat intake but positively with NSP. Females had greater diversity
Comments
General
Although this is well written and presented you studied just 60 subject when most similar studies use >500 subjects. Thus the study is very underpowered and one is left wondering how robust the results are.
Specific comments
1. Healthy volunteers were recruited from the faecal transplant clinic and comprised both donors and their relatives. Given the stringent criteria applied to fecal donors they are very likely to be different from the general population so I rather think this makes extrapolation to the general population uncertain.
2. Figure 3 is striking but how much of this is explained by falling Bifidobacteria numbers and which species? Can you relate this to changes in dairy intake .
3. Given strong effect of age you should include age in the analysis of adherence. I am assuming this was not possible because of small numbers. Should be included as limitation.
4. The gender differences are unexplained. The numbers are very small, 5/18 males with good adherence versus 12/21 females so one suspects the findings are not that robust especially factoring in the wide age range. It would be useful to have a table with the numbers by gender and age categories, I suspect some of the cells have very few subjects which should be mentioned under limitations.
5. Despite this it would be interesting to look at relevant gender diet differences ? red meat consumption or alcohol?
6. Figure 5 might be better reporting good adherence which is what we are really interested in versus medium rather than vice versa.
7. It would be good to try to explain why the Shannon index correlated inversely with fat intake and positively with non-starch polysaccharides (NSPs) and total sugars. You report a link between olive oil consumption and the abundance of Odoribacter, Clostridium XIVb, Victivallis, Bilophila, Dialister, and Phascolarctobacterium and negatively with Lactococcus, Faecalicoccus, Slackia, Clostridium sensu stricto, Romboutsia, and Collinsella. Can you identify any features these have in common that might explain this link?
8. Similarly with NSP , Glucose intake and Red meat intake there are many diverse species associated but is there a common theme?
9. Analysis of metabolic pathways in healthy subjects showed those with good adherence to the Mediterranean diet to be characterised by greater representation of two biosynthesis pathways, namely, the superpathway of (Kdo)2-lipid A biosynthesis and of ppGpp biosynthesis, and two toluene degradation pathways, namely, toluene degradation I (aerobic) (via o-cresol) and toluene degradation II (aerobic) (via 4-methylcatechol). Iy is not clear to me hoiw significant this is? Does it relate to any biomarkers of health?
10. Your analysis inferred SCFA production from the bacterial genes but it would have been better to measure fecal SCFAs combined with whole gut transit since without this data fecal SCFA is hard to interpret given accelerating transit increases SCFA concentrations possibly related to inadequate absorption.
Author Response
Dear Editor,
Please find enclosed our manuscript entitled “Impact of the Mediterranean Diet on the Gut Microbiome of a Well-Defined Cohort of Healthy Individuals.” to be reconsidered for publication in Nutrients as original article.
The authors have read the reviewers’ comments and made the necessary corrections.
Below, we provide a point-by-point reply to the reviewers’ comments.
Comments to Reviewer #1
Comments and Suggestions for Authors
This is a cross-sectional study of the microbiota in a group of Spanish fecal transplant donors and their relatives. It asked subjects to document their adherence to a mediterranean diet using the Med-DQI. Adherence was associated with greater alpha-diversity and increased counts of Paraprevotella and Bacteriodes. Alpha diversity was negatively correlated with fat intake but positively with NSP. Females had greater diversity
General
Although this is well written and presented you studied just 60 subject when most similar studies use >500 subjects. Thus the study is very underpowered and one is left wondering how robust the results are.
Response: Some of our results agree with those of other studies, which suggests that despite being a small cohort of individuals, we have obtained consistent results. Perhaps the fact that it is such a controlled cohort is what gives it greater robustness, given that in other cohorts, individuals have more variability and need a greater n to minimise changes in the intestinal microbiota due to other factors.
Specific comments
- Healthy volunteers were recruited from the faecal transplant clinic and comprised both donors and their relatives. Given the stringent criteria applied to fecal donors they are very likely to be different from the general population so I rather think this makes extrapolation to the general population uncertain.
Response: This poses an interesting issue, precisely the strict criteria applied adds value to our study since other confounding conditions that alter the microbiota are not present. In studies of this type, there are no fixed criteria for what constitutes a healthy individual. Some studies simply included individuals without digestive pathologies; however, there are other types of pathologies in which the intestinal microbiota is modified. The population in this study is a population with strict health criteria, hence our adjective healthy population. As the gut microbiota is affected by so many factors, we believed it was necessary to be able to describe possible effects of diet in a very specific cohort in order to minimise the effects of other factors as much as possible, because this is precisely one of the points that makes studies of the gut microbiota most difficult.
- Figure 3 is striking but how much of this is explained by falling Bifidobacteria numbers and which species? Can you relate this to changes in dairy intake.
Response: In our group of children, we included some breastfeeding babies, which clearly explains the change in the intestinal microbiota of this group. It has been described that because Bifidobacterium are more abundant in the breastfeeding stage and as other food groups are introduced, this group of bacteria or phylum decreases. The increase in Firmicutes and Prevotella is related to fiber and carbohydrate intake, and the increase in Bacteroidetes is related to the introduction of animal protein.
Thanks to this comment, we have included a small paragraph in the discussion of this change and added a description of the two major genera in each age group in the Results section.
We cannot answer the question of which species are altered because our analyses are at the genus level and not at the species level.
Also, thanks to this comment, we have added the following paragraph in the Results section (lines 261-265):
The most abundant genera in each age group were in Children: Bifidobacterium (27.2%) and Bacteroides (12%), in Teenager Bacteroides (11.36%) and Bifidobacterium (10. 33%); in Young adult Bacteroides (21.6%) and Faecalibacterium (7.3%); in Middle age Bacteroides (14.6%) and Prevotella (8.8%), and in Older adult Bacteroides (16.15%) and Blautia (7.6%).
And this other paragraph in the Discussion section (lines 407-411):
As we can observe in our data, there was a change in most phyla between children and other age groups. This is determined because in infants Bifidobacterium are abundant and as complementary feeding is introduced other microorganisms such as Firmicutes and Prevotella increase due to the introduction of foods rich in fiber and carbohydrates and Bacteroidetes due to the introduction of animal proteins [13].
- Given strong effect of age you should include age in the analysis of adherence. I am assuming this was not possible because of small numbers. Should be included as limitation.
Response: We agree with your suggestion, thanks to it, we have added the following paragraph in the Results section (lines 305-319):
When we compared subjects with good adherence to the Mediterranean diet within each age group with those with medium adherence, we found the following genera with significant changes in abundance: In the Teenager group, we found that individuals with good adherence had decreased numbers of Acidaminococcus, Cerasicoccus, Barnesiella, Butyricicoccus, Holdemanella, Paraprevotella, and Streptococcus. In the Young Adult group, Ruminococcus, Butyricicoccus, Clostridium IV, and Phascolarctobacterium were increased in the good adherence group. In the Middle Age group, more genera appeared, with significant changes according to diet type. Individuals with good adherence had fewer Succiniclastum, Methanobrevibacter, Paraprevotella, and Megasphaera. The abundance of the following genera increased: Mogibacterium, Alloprevotella, and Megamonas. We found several genera in which some OTUs increased and others decreased within the same genus, including Prevotella, Bacteroides, and Phascolarctobacterium. Finally, in Older Adults, we found that the only genus that increased with good adherence were Collinsella and one OTU of Prevotella, whereas Bacteroides and three OTUs of Prevotella, Coprococcus, Akkermansia, and Elusimicrobium decreased.
- The gender differences are unexplained. The numbers are very small, 5/18 males with good adherence versus 12/21 females so one suspects the findings are not that robust especially factoring in the wide age range. It would be useful to have a table with the numbers by gender and age categories, I suspect some of the cells have very few subjects which should be mentioned under limitations.
Response: We had already addressed the limited number of subjects in the paragraph referring to our study limitations. However, it was possible to make the following observations:
The median age of females in the good adherence group was 51 years and that in the medium adherence group 34 years (p=0.135). In males, the median age of the good adherence group was 26 years and that of the poor adherence group was 28 years (p=0.692). In the female group, we observed more differences in age between the diet adherence groups, but these differences were not significant. Furthermore, when we studied changes in the microbiota within these age groups, the genera that appear with significant changes by age do not coincide with those that appear with significant changes by sex, so we can conclude that the change in the bacterial genera that we found in the female group is not due to age. In the male group, it would be clearer as there was no change between the ages of the two adherence groups.
5. Despite this it would be interesting to look at relevant gender diet differences ? red meat consumption or alcohol?
Response: This is indeed an interesting point. We found no significant differences in the median MDQI score between males and females. However, we found that females consumed half as much red meat in grammes/day as males (20g/day for females; 40g/day for males; p=0.028). Otherwise, we found no significant differences between the two sexes.
- Figure 5 might be better reporting good adherence which is what we are really interested in versus medium rather than vice versa.
Response: We believe that the actual format is as representative as your suggestion. However, if finally the editorial office agrees on it we will adapt the figure.
- It would be good to try to explain why the Shannon index correlated inversely with fat intake and positively with non-starch polysaccharides (NSPs) and total sugars. You report a link between olive oil consumption and the abundance of Odoribacter, Clostridium XIVb, Victivallis, Bilophila, Dialister, and Phascolarctobacterium and negatively with Lactococcus, Faecalicoccus, Slackia, Clostridium sensu stricto, Romboutsia, and Collinsella. Can you identify any features these have in common that might explain this link?
Response: Thanks to this observation, we have added the following paragraph to the Discussion section (lines 455-469):
The positive correlation we found between Shannon’s index (alpha diversity) and NSPs was also observed by Martinez et al., who fed mice a diet rich in NSPs [20]. The effects of a Western diet rich in fats and simple sugars on gut microbiota usually involve a decrease in alpha diversity [31]. We found the same correlation with respect to fats but not with respect to total sugars. Note that our group of volunteers followed a Mediterranean diet, and none had poor adherence; therefore, sugar consumption was not very high in any case.
In relation to the correlation of some nutrients with specific genera, we can observe that in the study of olive oil consumption, we found a positive correlation with genera that have been seen to be found with a lower abundance in certain diseases such as inflammatory bowel disease, hypercholesterolemia, gastric cancer, or type II diabetes mellitus [28,32-37]. This suggests that these microorganisms are markers of good health status. In turn, the genera we found with a negative correlation with olive oil consumption were related to inflammatory bowel disease, immune-mediated diseases, and obesity [38-44]
- Similarly with NSP, Glucose intake and Red meat intake there are many diverse species associated but is there a common theme?
Response: Thanks to this comment, we have added the following paragraph to the Discussion section (lines 470-494):
In relation to the intake of NSPs, we found a positive correlation of genera such as Lactococcus, Victivallis, Anaerofustis, Senegalimasillia, and Phascolarctobacterium, which are related to SCFA production, good health status, and higher longevity [34,37,38,45]. However, Finegoldia and Peptoniphilus are also associated with inflammatory processes and the presence of conventional adenomas [46,47].
The genera Olsenella and Parvimonas were negatively correlated with NSPs and are associated with osteoporosis, markers of inflammation, ulcerative colitis, and colorectal cancer [48-50]. However, we also found a negative correlation with Dialister, which is decreased in patients with gastric cancer [36], or with Bifidobacterium, although in this case it has been shown that carbohydrates are essential for the colonisation of this genus and there is a great variety of species depending on the diet or the type of carbohydrates consumed [51]. We found a positive correlation of the genera Faecalicococcus, Clostridium sensu stricto, Butyricimonas, and Dialister with glucose levels. The first three have been related to inflammatory bowel disease and cirrhotic patients with hepatocellular carcinoma [38,52]; however, Dialister is decreased in patients with gastric cancer [36]. With respect to a glucose negative correlation, we found genera related to good health or even protective against some diseases such as pancreatic cancer [34,53], but we also found Olsenella, which is more related to an increase in diseases such as ulcerative colitis [49].
Finally, concerning the genera that correlated positively with red meat consumption, we found genera related to inflammatory bowel disease, such as Faecalicococcus and Clostridium sensu stricto [38,41]; Anaerofustis related to ulcerative colitis but also to greater longevity [38,54]; and other genera more related to a good state of health, such as Phascolarctobacterium and Subdoligranulum [37,55]. The genera found to be inversely correlated with red meat consumption were mostly related to anti-inflammatory function, protection against some types of cancer, or low risk of celiac disease [53,56,57].
- Analysis of metabolic pathways in healthy subjects showed those with good adherence to the Mediterranean diet to be characterised by greater representation of two biosynthesis pathways, namely, the superpathway of (Kdo)2-lipid A biosynthesis and of ppGpp biosynthesis, and two toluene degradation pathways, namely, toluene degradation I (aerobic) (via o-cresol) and toluene degradation II (aerobic) (via 4-methylcatechol). Iy is not clear to me hoiw significant this is? Does it relate to any biomarkers of health?
Response: The biosynthesis pathways are not related to any biomarker of health, but as mentioned in the discussion, they have been related to several essential bacterial functions as well as in their adaptation and resistance processes, which could be related to a gut microbiota more resistant to dysbiosis or better adapted to changes. Toluene degradation pathways are related to the transformation of toxic aromatic hydrocarbons into other compounds that are less toxic to the host and produce less inflammation.
We have added this information to the Discussion section (lines 501-503):
Toluene degradation pathways are related to the transformation of toxic aromatic hydrocarbons into other compounds that are less toxic to the host and cause less inflammation [58].
- Your analysis inferred SCFA production from the bacterial genes but it would have been better to measure fecal SCFAs combined with whole gut transit since without this data fecal SCFA is hard to interpret given accelerating transit increases SCFA concentrations possibly related to inadequate absorption.
Response: We fully agree, it is always better to measure faecal SCFA but for this study we were not able to do so. However, it should be noted that our volunteers did not suffer from accelerated transit.

Reviewer 2 Report
Comments and Suggestions for Authors
Cuesta et. al., in their current study, investigated the effect of Mediterranean diet on the gut microbiome of healthy cohorts. In their study, they enrolled 60 healthy volunteers, including children, adults, middle-aged adults and older adults. However, the discussion part is poorly written. The study has lots of useful information, however, there is no proper correlation between all the information and lacks to deliver the importance. I have the following comments:
1. The introduction needs to be more detailed including the importance of various biological pathways on the individual’s health and what is the effect of poor diet on them.
2. The authors, did not mention the duration of the study. How long did the individuals adhered to the Mediterranean diet before their stools were collected? Was there only one-time point or different time points?
3. The discussion needs to be rewritten with detailed explanations to the results obtained. The authors have attempted to analyze the composition of gut microbiota, acknowledging Firmicutes as the most abundant phylum in both males and females of all age group. However, they did not discuss this in details in their discussion and the detailed explanation to how it affects the health or related pathways.
4. In the results, the authors found various underlying pathways such as enzymes involved in lipid metabolism, enzymes in secondary metabolite biosynthesis, however, what’s the importance of these pathways and how it affects the health of any individual is lacking.
5. The conclusion is poorly written and need to include important observations found in the results section.
6. Also, the plagiarism report shows a lot if similarity in the study designing and should be changed to make it more impactful and not merely just a copy of other authors study design.
Author Response
Comments to Reviewer #2
Comments and Suggestions for Authors
Cuesta et. al., in their current study, investigated the effect of Mediterranean diet on the gut microbiome of healthy cohorts. In their study, they enrolled 60 healthy volunteers, including children, adults, middle-aged adults and older adults. However, the discussion part is poorly written. The study has lots of useful information, however, there is no proper correlation between all the information and lacks to deliver the importance. I have the following comments:
- The introduction needs to be more detailed including the importance of various biological pathways on the individual’s health and what is the effect of poor diet on them.
Response: Thanks to this suggestion, we have added the following paragraphs to the introduction (lines 64-102):
Diets rich in fats and refined carbohydrates (typical Western diets) have become more widely accepted, obesity and related metabolic disorders have become highly prevalent and are increasing significantly [1,2]. Components of the metabolic syndrome, such as hypertriglyceridaemia, low serum HDL-cholesterol levels, and insulin resistance, increase the risk of developing diabetes, cardiovascular disease and hepatic steatosis [3,4]. Intestinal bacteria play an important role in the synthesis and absorption of various nutrients and metabolites, including lipids, bile acids, short-chain fatty acids (SCFA), amino acids, and vitamins.
The macronutrients that make up the human diet are carbohydrates, proteins, and fats. Carbohydrates act as a source of energy, help control blood glucose and insulin metabolism, and participate in the metabolism of cholesterol and triglycerides [5]. An inadequate amount of carbohydrates in the diet can affect various physiological and metabolic processes. Highly processed foods rich in simple carbohydrates are less nutritious and cause a sharp rise in blood glucose. The healthiest sources of complex carbohydrates are legumes, vegetables, or whole grains because of their reduced effects on blood glucose [5]. Obesity is related to the excessive consumption of simple carbohydrates, and this condition can have consequences such as increased risk of cardiovascular disease, or type 2 diabetes. However, foods with a low glycaemic index and rich in non-starch polysaccharides protect against diabetes [5].
Dietary proteins interact with the physiological functions of the gut to control amino acid, glucose, and lipid metabolism [6]. The effects of dietary proteins and glucose metabolism are well established. When consumed with carbohydrates, dietary proteins reduce glycemic responses [7]. Some metabolic disorders such as kwashiorkor and marasmus are caused by severe protein deficiencies [8]. Proteins are fundamental building blocks of tissues in humans. Thus, amino acids are essential for the health, development, reproduction, and survival of organisms. In the elderly, less severe forms of dietary protein deficiency occurs, increasing susceptibility to metabolic diseases [8]. On the other hand, excessive protein intake can compromise health, especially in those with liver or kidney dysfunction [9].
The quality and quantity of dietary lipids play a key role in metabolic and cardiovascular health [10]. An excessive intake of long-chain saturated fatty acids together with reduced intake of unsaturated fatty acids (mainly omega-3), represents one of the main drivers of metabolic disorders attributed to the Western diet, obesity and its cardiometabolic complications [10]. In some tissues such as skeletal muscle or liver, long-chain saturated fatty acids trigger metabolic inflammation, mitochondrial dysfunction which promotes metabolic derangements [10,11]. In particular, the effects of long-chain saturated fatty acids on metabolic health appear to be underpinned by lipotoxicity, which was proposed as a key phenomenon linking lipid metabolism to obesity and its comorbidities [12].
- The authors, did not mention the duration of the study. How long did the individuals adhered to the Mediterranean diet before their stools were collected? Was there only one-time point or different time points?
Response: Participants in this observational study, have the Mediterranean diet as their regular base diet, this study is conducted in Spain where the Mediterranean diet is the most widespread diet. Due to the observational nature of this study, none of the participants were subjected to any dietary habit change, the dietary intake questionnaire was referred to their diet in the last month before the sample was collected.
Regarding the changes over time in the microbiota of these subjects, we have analysed in our cohort of donors this phenomenon. We haven´t observed any significant changes in the microbiota of the subjects in any of the different time points evaluated. We have included for the purpose of this study the first sample of each subject.
- The discussion needs to be rewritten with detailed explanations to the results obtained. The authors have attempted to analyze the composition of gut microbiota, acknowledging Firmicutes as the most abundant phylum in both males and females of all age group. However, they did not discuss this in details in their discussion and the detailed explanation to how it affects the health or related pathways.
Response: Thanks to this suggestion, we have included a more detailed explanation of our results in the Discussion section (lines 412-436):
Within Firmicutes, the most common families in our volunteers were Lachnospiraceae and Ruminococcaceae. These Firmicutes families hydrolyse starch and other sugars and produce butyrate and other SCFA [14]. SCFA activity modulates the microbial environment and directly interacts with the host immune system [15]. In addition, SCFA lead to changes in glycolysis and fatty acid metabolism in colonic epithelial cells and a decrease in inflammatory markers [16]. Within Firmicutes, elevated abundance of Lachnospiraceae was positively correlated with glucose and/or lipid metabolism [17,18].
Certain species respond to certain dietary carbohydrate changes, mainly bacteria specializing in using resistant starch or non-starch polysaccharides (NSP). Some members of the Roseburia group were major responders to diets enriched in resistant starch [19]. Some studies have shown that a diet rich in whole-grain cereals increases the alpha diversity and abundance of Firmicutes [20]. Other Lachnospiraceae species are strongly influenced by NSP-rich diets [16]. However, some genera of Lachnospiraceae actively alter glucose metabolism, leading to inflammation and promoting the development of both type 1 and type 2 diabetes [21-23]. Different species of Lachnospiraceae have been associated with impaired lipid metabolism and thus obesity. It should not be forgotten that certain dietary fats, such as omega-3 polyunsaturated fatty acids, can reduce the risk of death from coronary heart disease and the development of breast cancer [24,25]. In animal models, a diet enriched with omega-3 has been shown to increase the abundance of Lachnospiraceae [26].
In contrast, the Ruminococcaceae family is strictly anaerobic and present in the colonic mucosa of healthy individuals [27]. In inflammatory bowel diseases, hepatic encephalopathy, and Clostridioides difficile infection, a decreased abundance of Ruminococcaceae has been observed [28-30]. This family produces butyrate and other SCFAs and therefore plays an important role in maintaining gut health.
- In the results, the authors found various underlying pathways such as enzymes involved in lipid metabolism, enzymes in secondary metabolite biosynthesis, however, what’s the importance of these pathways and how it affects the health of any individual is lacking.
Response: Thanks to this statement, we have added this paragraph to the Discussion section (lines 504-513):
We found a greater representation of enzymes involved in lipid metabolism in individuals with good adherence to the Mediterranean diet. It is well known that the metabolism of complex lipids by the intestinal microbiota modulates the lipid homeostasis of the host, and that a lipid imbalance can have important consequences on health [59], to date the exact relationship of these particular pathways have not yet been explored. Other enzymes linked to Mediterranean diet adherence in our study, have been found to be related to the metabolism of L-tyrosine, which is a precursor of some neurotransmitters such as adrenaline and dopamine and have important effects on behavior or mood [60]. Owing to the importance of this amino acid, its depletion in the diet has been associated with an increased risk of clinical depression [60].
- The conclusion is poorly written and need to include important observations found in the results section.
Response: We have revised the conclusion, and have rewritten it as follows (lines 525-542):
In conclusion, our study provides helpful insights into the complex relationship between the Mediterranean diet and intestinal microbiome. Thorough classification by age and sex, along with the integration of functional metagenomics, distinguishes our research from the existing literature. In this study, we observed the evolution of the gut microbiota of healthy individuals with age and were able to distinguish changes in this microbiota within a Mediterranean diet based on adherence. Thus, we found that individuals with good adherence to the Mediterranean diet had a higher abundance of genera such as Paraprevotella and Bacteroides, which are associated with good health. We also observed a correlation between certain health-promoting foods, such as olive oil or fiber consumption, and certain bacterial genera that are related to the synthesis of SCFAs and the absence of disease. Within the Mediterranean diet, we observed certain foods, such as red meat, associated with microorganisms that are less beneficial to health.
The positive correlations between adherence and microbial diversity and the identification of specific taxa and functional pathways provide further insight. These findings not only affirm the existing knowledge but also open avenues for extending research into sex-specific responses and the functional implications of dietary patterns on the gut microbiome. Ultimately, our study adds depth to the understanding of how dietary choices shape the microbial landscape and consequently influence human health.
- Also, the plagiarism report shows a lot if similarity in the study designing and should be changed to make it more impactful and not merely just a copy of other authors study design.
Response: We regret that our responses to the plagiarism report did not reach the reviewers to avoid this misunderstanding. The methods section appears to have a high percentage of plagiarism because it has indeed detected two articles from our group published prior to this one. The methodology followed by our center has not changed during this time, therefore the center, the criteria used for healthy individuals, sample collection, sample processing and the generation of libraries for sequencing are the same. We would like to clarify that we have not copied the study design from other authors, the plagiarism report detected our previous publications. We have tried to vary the wording of the section to comply with the plagiarisms journal´s requirements.
In addition, the plagiarism report showed many repetitions of expressions or terms typical of this type of study such as “Mediterranean diet” or “gut microbiota”, which are obviously repeated in many publications.

Round 2
Reviewer 1 Report
Comments and Suggestions for Authors
Thank you for your responses
Unfortunately without more participants you cannot be sure that the differences you observed are robust but you do discuss this in your discussion of limitations so I think this is reasonable
Author Response
I fully agree that the results would be more robust if we had more subjects. Thank you very much for your comments and input.
Reviewer 2 Report
Comments and Suggestions for Authors
The authors have read the reviewers’ comments and made the necessary corrections and hence now can be considered for publication
Author Response
We appreciate your comments and input.